# Thiamine hydrochloride, riboflavin, pyridoxine hydrochloride, and biotin hard gelatin capsules prepared in advance and stored for the treatment of pediatric metabolic diseases: a safer alternative

**Marine Bonino**[1], **Amélie Bouchez**[1], **Amna Kadri**[1], **Ines Jacquet**[1],
**Romain Paoli-Lombardo**[1,2], **Christophe Jean**[3], **Mélanie Fuchs**[4], **Maeva Montaleytang**[4],
**Patrice Vanelle**[1,2], **Thierry Terme**[2], **Pascal Rathelot**[1,2], **Christophe Curti**[1,2]*

1 Service central de la qualité et de l'information pharmaceutiques (SCQIP), Pharmacy Department, AP-HM, Marseille, France, 2 CNRS, Institut de Chimie Radicalaire ICR, UMR, Equipe de Pharmaco-Chimie Radicalaire, Aix Marseille Univ, Marseille, France, 3 Pharmacie Sainte Marguerite, Pharmacy Department, AP-HM, Marseille, France, 4 Pharmacie Timone, Pharmacy Department, AP-HM, Marseille, France

* christophe.curti@ap-hm.fr, christophe.curti@univ-amu.fr

## Abstract

The treatment of several pediatric metabolic diseases involves vitamins supplementation. Among these, thiamine, riboflavin, pyridoxine and biotin can be prescribed and compounded as hard gelatin capsules. In compounding practice, a medication can be done extemporaneously, leading to a risk of error. However, a medication can also be done in advance, analytically controlled and stored. Such practice reduce the risk of error and decrease the cost, but also imposes the realization of stability studies to establish beyond-use-dates. Thiamine hydrochloride, riboflavin, pyridoxine hydrochloride, and biotin hard gelatin capsules chromatographic and microbiological methods were both validated and used to perform stability studies. Thiamine hydrochloride 50 mg hard gelatin capsules with microcrystalline cellulose and silica as excipients are stable for 6 months when stored at 25 °C/ 60% RH protected from light. Riboflavin 50 mg with microcrystalline cellulose, pyridoxine hydrochloride 50 mg with microcrystalline cellulose and biotin 40 mg with microcrystalline cellulose/silica are stable for one year when stored at 25 °C/ 60% RH protected from light. These results allow the compounding in advance of batches of 300 capsules controlled, stored, and quickly dispensed in case of an emergency, such decreasing the risk of error and/or iatrogenic event.

## 1. Introduction

In pediatric practice, vitamins B1 (thiamine), B2 (riboflavin), B6 (pyridoxine), or B8 (biotin) can be prescribed for neonatal metabolic disorders [1–2]. While the parenteral route is

**Data availability statement:** All relevant data are within the manuscript and its Supporting Information files.

**Funding:** The author(s) received no specific funding for this work.

**Competing interests:** The authors have declared that no competing interests exist.

commonly used for emergency treatment with thiamine [3] and pyridoxine [4], the oral route should be prioritized as soon as feasible [1–4].

The thiamine dose for oral supplementation for the treatment of thiamine deficiency disorders can vary widely, from, for example 50 mg to 1 500 mg per day, according to the indication and the severity of the illness [3,5]. Riboflavin can be orally prescribed in several indications, but high-dose between 10 mg.kg$^{-1}$ and 50 mg.kg$^{-1}$ per day can be administered in a rare neurological condition named riboflavin transporter deficiency [6]. Pyridoxine can be administered *per os* for the treatment of pyridoxine-dependent epilepsy at 15–30 mg.kg$^{-1}$ per day divided into 3 doses in infants and up to 200 mg.day$^{-1}$ in neonates [7–9]. In biotinidase and holocarboxylase synthetase deficiencies, biotin is administered *per os* at dosages ranging from 5 mg.day$^{-1}$ to 40 mg.day$^{-1}$ [10–12].

To ensure these prescriptions, several commercial specialties are available in our country (France) as summarized in Table 1. However, some available commercial formulations can have content not compatible with pediatric dosage regimen, or can contain excipients with known effects (as defined and listed by the European Medicine Agency [13]). In pediatric practice, some prescribers use parenteral form administered orally to reach the required dosage, but this practice must be avoided. Indeed, parenteral available commercial formulations can have high osmolarities which should be problematic for neonate's treatment [14].

Therefore, these treatments must be compounded by the hospital pharmacists. In pediatric practice, liquid and solid oral formulations both have advantages and inconveniences. Solid forms do not need to be swallowed whole and should be dissolved in a small volume of liquid before administration. The advantages of hard gelatin capsules over oral liquids include improved stability, good dosage uniformity, lower cost and simplicity of their formulation. This simplicity also contributes to better safety, for instance through the absence of any preservatives, and improves the access to age-appropriate essential medicines for young children, particularly in low- and middle-income countries [15–16].

For the treatment of pediatric metabolic diseases, hard capsules can be compounded: thiamine hydrochloride 50 mg, riboflavin 50 mg, pyridoxine hydrochloride 50 mg, and biotin 40 mg capsules. Such prescriptions can be received during the weekend or night duty by a pharmacist intern. Emergency compounding is never a good solution because it can cause disorganization, delay, or even error, yielding to an iatrogenic event. Therefore, we decided to realize a stock of each of these capsules and subsequent stability studies were performed. We present herein the results of our work, which can allow every pharmacist to compound similar storable, harmonized, and stable formulas.

**Table 1. B1, B2, B6, and B8 vitamins available commercial formulations and required oral unidose formulas.**

| Vitamin | Commercial drugs | Excipients with known effects |
| --- | --- | --- |
| B1 (thiamine hydrochloride) | 250 mg tablets | Sucrose |
| | 100 mg.mL$^{-1}$ IM solution | Phenol |
| | 50 mg.mL$^{-1}$ IM/IV solution | |
| B2 (riboflavin) | None with riboflavin alone | |
| B6 (pyridoxine hydrochloride) | 250 mg breakable tablets | |
| | 50 mg.mL$^{-1}$ IM/IV solution | Sodium sulfite in certain specialties |
| B8 (biotin) | 5 mg tablets | Lactose |
| | 5 mg.mL$^{-1}$ IM solution | |

## 2. Materials and methods

### 2.1. Capsules compounding

To determine the formula for each vitamin capsule, we first evaluated the bulk and tapped densities for each individual component and for selected mixtures, using the same protocol as previously described [17]. Then, we calculated the theoretical weights to achieve a volume of 63 mL (300 hard gelatin capsules, size 4) with the appropriate amount of Active Pharmaceutical Ingredient (API). Finally, we slightly rounded these theoretical weights to obtain a simplified formula, ensuring differences between the theoretical weight and the rounded weight lower than 1%. The results are summarized in a table in the supporting information (S1 Table).

For thiamine hydrochloride 50 mg hard capsules compounding, we first tried a volume-based protocol. Then, as explained in the results, a second weight-based method was implemented, with adjunction of silica. Compounding formulas are detailed in Table 2. Except for these differences, hard gelatin vitamin capsules were realized under the same protocol. Pharmaceutical-grade API (thiamine hydrochloride or riboflavine or pyridoxine hydrochloride or biotin (Cooper France)) and excipients (microcrystalline cellulose (Cooper France), silica (Aerosil® 200, Inresa France)) were weighed on a qualified precision balance (Precisa XT220A, Precisa). One spatula tip (approximately 2 mg) of red carmine (Fagron France) was added as a homogenization tracer. The mixture was transferred to a mortar for gentle mixing with a pestle. Three hundred hard gelatin capsules (size 4, ivory) were placed on a manual capsule-filling machine (ProFiller 3700, LGA). The caps were separated from the empty bodies, and the entire mixture was inserted into the capsule bodies. Finally, bodies and caps were sealed.

### 2.2. HPLC stability-indicating dosing methods

General parameters and vitamin capsules treatment before analysis are summarized in Tables 3–4.

The thiamine hydrochloride dosing method was adapted from the method described in the European Pharmacopoeia ("Thiamine hydrochloride" monograph) with slight variations in the mobile phase and the gradient. It was performed with a Kromasil® C18, 4.6 × 250 mm 5 µm column (AIT France) heated at 45 °C. Separation was attained with mobile phase A (3.764 g.L$^{-1}$ solution of sodium heptanesulfonate (MP Biomedicals) in ultrapure water) and mobile phase B (methanol (VWR)). To quantify thiamine hydrochloride in a hard capsule, 20 mg of a certified reference standard (PHR1037, Sigma-Aldrich, previously checked with primary standard reference material 1656002, Sigma-Aldrich) were weighed on a precision

**Table 2. Compounding formulas for 300 hard gelatin vitamin capsules.**

|  | Thiamine hydrochloride protocol 1 | Thiamine hydrochloride protocol 2 | Riboflavin | Pyridoxine hydrochloride | Biotin |
|---|---|---|---|---|---|
| Content per capsule | 50 mg | 50 mg | 50 mg | 50 mg | 40 mg |
| API weight | 15.0 g | 15.0 g | 15.0 g | 15.0 g | 12.0 g |
| MCC[a] | 111 mL | 5.0 g | 15.0 g | 12.0 g | 12.5 g |
| Silica | – | 0.75 g | – | – | 0.75 g |
| Red carmine | yes[b] | yes[b] | no | yes[b] | yes[b] |
| Mixing duration | 25 min | 25 min | 10 min | 10 min | 10 min |

[a] Microcrystalline cellulose

[b] One spatula tip, approximately 2 mg

balance (Sartorius Cubis MCA 125P) and dissolved in 20 mL of mobile phase A. 250 μL are aliquoted and mixed with 750 μL of mobile phase A to obtain a concentration of thiamine hydrochloride at 250 μg.mL$^{-1}$.

The riboflavin dosing method was developed based on the European Pharmacopoeia ("riboflavin" monograph) with significant modifications in the mobile phase and the mode (from gradient to isocratic). It involved a Stability® Basic C18, 4.6 × 250 mm 5 μm column (Precision Instruments France) and a mobile phase made from a mixture of 80% of 1.3205 g.L$^{-1}$ of dibasic ammonium phosphate (Sigma-Aldrich) in ultrapure water, 1 mL of 85% phosphoric acid (VWR Chemicals) and 20% of acetonitrile (Fisher Chemical). To quantify riboflavin in a hard capsule, 50 mg of a certified reference standard (PHR1054, Sigma-Aldrich, previously checked with primary standard reference material R0600000, Sigma-Aldrich) were weighed on a precision balance and treated as a capsule to obtain a theoretical concentration of riboflavin at 150 μg.mL$^{-1}$.

The pyridoxine dosing method was developed from the United States Pharmacopoeia ("pyridoxine tablets" monograph, dissolution test) with modifications in the column specifications, the injection volume and the flow. A Nucleosil® C18, 4.6 × 250 mm 5 μm column (Macherey Nagel) was used with a mobile phase made with 600 mg of sodium hexanesulfonate (Sigma-Aldrich) dissolved in 700 mL of ultrapure water completed with 10 mL of glacial acetic acid (VWR Chemicals) and with a pH adjusted at 3.0 with NaOH 1M. Then, 235 mL of methanol was added, and finally, ultrapure water qs 1 L. To quantify pyridoxine hydrochloride in a hard capsule, 50 mg of a certified reference standard (PHR1036, Sigma-Aldrich, previously checked with primary standard reference material P4100000, Sigma-Aldrich) were weighed on a precision balance and dissolved in 100 mL of mobile phase to obtain a theoretical concentration of pyridoxine hydrochloride at 500 μg.mL$^{-1}$.

The biotin dosing method developed from the United States Pharmacopoeia ("biotin capsules" monograph) with significant modifications in the column specifications and injection volume. It was performed with an Eclipse® XDB-C8, 4.6 × 150 mm, 5 μm column (Agilent) and a mobile phase composed of a mixture of 91.5% of 1 g.L$^{-1}$ sodium perchlorate monohydrate (VWR Chemicals), 1 mL.L$^{-1}$ of 85% phosphoric acid (VWR Chemicals) in ultrapure water and 8.5% of acetonitrile. To quantify biotin in a hard capsule, 20 mg of a certified reference standard (PHR1233, Sigma-Aldrich, previously checked with primary standard reference material B1116000, Sigma-Aldrich) were weighed on a precision balance and dissolved in 10 mL of DMSO. The mobile phase was added qs 50 mL. 625 μL were aliquoted and mixed with 375 μL of a diluent (acetonitrile/ultrapure water (20/80)) to obtain a theoretical concentration of biotin at 250 μg.mL$^{-1}$.

Each method was validated to be stability-indicating. First, linearity was determined with at least five different concentrations prepared in quintuplicate from the secondary standard. Then, repeatability was determined from within-day variation measurements (at least n = 15), whereas intermediate precision and accuracy were determined from between-day variation measurements (at least n = 18). For each vitamin, three concentration levels were studied. Finally, forced degradation studies were performed to ensure the stability-indicating character of each method.

Forced degradation experiments were conducted with a comparison of standard solutions of thiamine hydrochloride (250 μg.mL$^{-1}$), riboflavin (150 μg.mL$^{-1}$), pyridoxine hydrochloride (500 μg.mL$^{-1}$), and biotin (250 μg.mL$^{-1}$). The studied experimental conditions are detailed in the results part for each molecule.

## 2.3. Disintegration tests

Disintegration tests were performed with an Agilent 100 automated disintegration apparatus. Briefly, one capsule was placed in each of six tubes in a 2-basket rack assembly, and the

**Table 3. HPLC stability-indicating dosing methods, general parameters.**

|  | Thiamine hydrochloride | Riboflavin | Pyridoxine hydrochloride | Biotin |
|---|---|---|---|---|
| Flow rate | 1 mL.min$^{-1}$ | 1 mL.min$^{-1}$ | 1.5 mL.min$^{-1}$ | 1.2 mL.min$^{-1}$ |
| Mode | Gradient[a] | Isocratic | Isocratic | Isocratic |
| Wavelength | 248 nm | 267 nm | 280 nm | 200 nm |
| Injection volume | 25 µL | 10 µL | 20 µL | 10 µL |
| Run duration | 30 min | 25 min | 15 min | 18 min |

[a]60% A at 0 min, from 60% to 55% between 0 and 12 min and 55% between 12 and 30 min

**Table 4. Vitamin capsules treatment before HPLC-UV analysis.**

|  | Thiamine hydrochloride 50 mg | Riboflavin 50 mg | Pyridoxine hydrochloride 50 mg | Biotin 40 mg |
|---|---|---|---|---|
| Capsule initial treatment | phase A (20 mL) | Diluent A[a] (25 mL) | DMSO (20 mL) | DMSO (20 mL) |
| Homogenization | vortexed and centrifuged (5 min, 3,000 rpm) | | | |
| Volume aliquoted | 1 mL | 300 µL | 10 mL | 100 µL |
| Final dilution | Phase A (9 mL) | Diluent B (3 700 µL) | Mobile phase (40 mL) | Mobile phase (700 µL) |
| Theoretical concentration | 250 µg.mL$^{-1}$ | 150 µg.mL$^{-1}$ | 500 µg.mL$^{-1}$ | 250 µg.mL$^{-1}$ |

[a]5 mL of NaOH 0.1 M (VWR Chemicals), qs 25 mL with a 13.6 mg.L$^{-1}$ solution of sodium acetate (Merck). [b] 13.6 mg.L$^{-1}$ solution of sodium acetate (Merck)

apparatus was operated with water (distilled water, Cooper) maintained at 37 ± 2 °C as immersion fluid and without disks. After 30 min, tubes were observed to determine if the capsules were completely disintegrated.

## 2.4. Microbiological examination for nonsterile products, suitability of the method and routine analyses (TAMC and TYMC)

The Total Aerobic Microbial Count (TAMC) and Total Yeast and Mold Count (TYMC) of vitamin capsules were done by the surface-spread method. Suitability of TAMC and TYMC was performed individually for each hard gelatin vitamin capsule (thiamine hydrochloride 50 mg capsules, riboflavin 50 mg capsules, pyridoxine hydrochloride 50 mg capsules and biotin 40 mg capsules) with the following protocol.

Under a microbiological safety cabinet (Herasafe KS, Thermo Scientific), five reference strains (Biomérieux Bioball: *Staphylococcus aureus* NCTC10788, *Bacillus subtilis* NCTC10400, *Pseudomonas aeruginosa* NCTC12924, *Candida albicans* NCPF3179, and *Aspergillus braziliensis* NCPF2275) were suspended in pH 7.2 sterile phosphate buffer to obtain 100–1,000 CFU.mL$^{-1}$. 100 µL (10–100 CFU) of each strain suspension were aliquoted in five tubes containing 5 mL of pharmacopeia diluent/ tween 80 (Dominique Dutcher). 100 µL (10–100 CFU) of each strain suspension were aliquoted in five tubes containing 5 mL of pharmacopeia diluent/ tween 80 and one capsule. One tube was prepared as a negative control with 5 mL of pharmacopeia diluent/ tween 80 and one capsule. Each tube was vortexed for at least 10 min. After 10 min, the capsule content was suspended in the diluent. 4 × 500 µL were aliquoted from each tube to inoculate two tryptic soy agars and two Sabouraud agars. Tryptic soy agars were incubated for 3 days in an incubator at 30–35 °C and Sabouraud agars for 5 days at 20–25 °C (Heratherm and Heraeus Thermo Scientific). At the end of the incubation, for each reference strain, the mean number of CFU of "vitamin capsule + reference strain" samples did not

vary by more than a factor of two from the mean number of CFU of "reference strain alone" samples, which confirms the suitability of this method for the microbiological examination of vitamin capsules.

Routinely and for stability studies, each analyzed capsule was preliminarily weighed. Under a microbiological safety cabinet, capsules were aliquoted in five tubes containing 5 mL of pharmacopeia diluent/ tween 80. One tube was prepared as a negative control with 5 mL of pharmacopeia diluent/ tween 80. Each tube was vortexed for 10 min. After 10 min, the capsule content was suspended in the diluent. $4 \times 500$ µL were aliquoted from each tube to inoculate two tryptic soy agars and two Sabouraud agars per capsule. Tryptic soy agars were incubated for 3 days in an incubator at 30–35 ˚C and Sabouraud agars for 5 days at 20–25 ˚C.

Mean CFU values were determined from the two tryptic soy agars CFU counts (TAMC) and from the two Sabouraud agars CFU counts (TYMC). Results in CFU.g$^{-1}$ were determined from mean CFU values, the weight of the analyzed capsule, and the dilution factor. Conformity was established if TAMC was found to be lower than 2,000 CFU.g$^{-1}$ and if TYMC was found to be lower than 200 CFU.g$^{-1}$.

## 2.5. Stability study

3 independent batches of each hard gelatin vitamin capsule (thiamine hydrochloride 50 mg capsules, riboflavin 50 mg capsules, pyridoxine hydrochloride 50 mg capsules, and biotin 40 mg capsules) were submitted to a stability study for 12 months, protected from light and stored in an environmental test chamber (25 ˚C, 60% RH (Relative Humidity)) (ICH110, Memmert).

The vitamin content and degradation products were evaluated at time zero, one week, one month, 2 months, 3 months, 4 months, 6 months, and one year after compounding. At each sampling point, 3 capsules of the first batch, 3 capsules of the second batch, and 3 capsules of the third batch were analyzed. Disintegration tests were performed on 6 capsules at time zero, one month, 3 months, 6 months, and 12 months. At the beginning and the end of the study, the three batches were analyzed for microbiological contamination (TAMC and TYMC), but for the 6 other intermediary sampling points (one week, one month, 2 months, 3 months, 4 months, and 6 months), only 2 capsules from the same batch were analyzed. Significant changes in vitamin content were defined as a 10% change in dosage from its initial value as previously reported for hospital-compounded preparations [18–21].

For thiamine hydrochloride 50 mg capsules made with protocol 2, a simplified stability study was performed under the same protocol but with only 4 sampling points (time zero, 3 months, 6 months, and 12 months).

The statistical analysis of the change in measured content over time was conducted in accordance with the ICH Q1E guidelines. Specifically, batch poolability was confirmed through an analysis of covariance (ANCOVA) using a significance level of 0.25. Since time × batch interactions and the main effect of batch were non-significant for each of the formulations studied, all batches were deemed poolable. Then, the relationship between the variation in active ingredient content over 12 months and time was estimated for each formulation using regression analysis and the calculation of the coefficient of determination R$^2$.

## 3. Results

Linearity was demonstrated for all four methods: between 50 and 500 µg.mL$^{-1}$ for thiamine hydrochloride, between 100 and 200 µg.mL$^{-1}$ for riboflavin, between 200 and 800 µg.mL$^{-1}$ for pyridoxine hydrochloride and between 100 and 500 µg.mL$^{-1}$ for biotin. Repeatability, intermediate precision, and accuracy results are detailed in Table 5.

**Table 5. Thiamine, riboflavine, pyridoxine and biotine method validation.**

| Samples (concentration levels, µg.mL⁻¹) | % RSD within-day | | | % RSD between-day | | | Bias | | |
|---|---|---|---|---|---|---|---|---|---|
| Thiamine chlorhydrate (200/ 250/ 300) | 0.29% | 0.29% | 0.31% | 2.08% | 2.48% | 2.22% | 0.30% | 0.01% | 0.77% |
| Riboflavine (135/ 150/ 165) | 0.43% | 1.30% | 0.68% | 1.29% | 2.74% | 2.18% | 1.50% | 1.00% | 2.10% |
| Pyridoxine chlorhydrate (400/ 500/ 600) | 0.76% | 0.42% | 0.46% | 3.37% | 2.96% | 3.69% | 1.61% | 0.57% | 1.27% |
| Biotine (225/ 250/ 275) | 0.78% | 1.87% | 2.18% | 2.00% | 1.25% | 1.18% | 1.58% | 1.22% | 1.57% |

Results from forced degradation studies are reported in Tables 6–9 and resulting chromatograms can be found in supporting information (S1-S4 Figs). The lower values reported for resolution were respectively 1.79 (peak with RRT 0.95, thiamine chlorhydrate), 2.31 (peak with RRT 1.07, riboflavin), 1.41 (peak with RRT 0.91, pyridoxine chlorhydrate) and 2.84 (peak with RRT 1.10, biotin). The four methods have good specificity, because all resolution values for degradation products were higher than 1.30 [22].

Moreover, a System Suitability Test (SST) was proposed for each dosing method. The limit values were based on (in order of priority, if available): the USP monograph of the corresponding pharmaceutical drug [23–24], the USP monograph of the corresponding Active Pharmaceutical Ingredient [25–26] or the FDA recommendations [27]. Proposed values are reported in Table 10.

**Table 6. Thiamine chlorhydrate forced degradation.**

| Conditions | % degradation | Degradation products RRT |
|---|---|---|
| 80 °C, 10 days | 30% | 0.21; 0.25; 0.32; 1.09; 1.15; 1.23; 1.32 |
| $H_2O_2$ 1.5%, 10 days | 14% | 0.14; 0.25; 0.32; 0.35; 0.37; 0.43; 0.51; 0.67; 0.95; 1.09 |
| HCl 2 N, 20 days | 12% | 0.32 |
| NaOH 0.5 M, 45 min | 17% | 0.32; 0.47; 1.09 |
| Light 20 days | 0% | |

**Table 7. Riboflavine forced degradation.**

| Conditions | % degradation | Degradation products RRT |
|---|---|---|
| 50 °C, 22 h | 60% | 0.47; 0.71; 0.90; 1.07; 1.22; 1.28; 1.37; 1.51 |
| $H_2O_2$ 15%, 40 h | 0% | – |
| HCl 2 N, 24 h | 3% | 1.82 |
| NaOH 1 M, 19 h | 76% | 1.37; 1.82 |
| Light 22 h | 100% | 0.47; 0.71; 0.90; 1.07; 1.22; 1.28; 1,51; 2.00; 3.76 |
| Light 20 min | 64% | 0.47; 0.71; 1.22; 1.51; 1.70; 2.00; 3.76 |

**Table 8. Pyridoxine chlorhydrate forced degradation.**

| Conditions | % degradation | Degradation products RRT |
|---|---|---|
| 80 °C, 96 h | 2% | 1.42; 2.96 |
| $H_2O_2$ 15%, 96 h | 11% | 2.70 |
| HCl 2 N, 96 h | 0% | – |
| NaOH 3 N, 96 h | 4% | 0.49 |
| Light, 96 h | 35% | 0.53; 0.60; 0.75; 0.91; 1.42 |

**Table 9. Biotin forced degradation.**

| Conditions | % degradation | Degradation products RRT |
|---|---|---|
| 80 °C, 48 h | 9% | 0.10 |
| $H_2O_2$ 3%, 1 h | 98% | More than 15 peaks, with 0.16; 0.50; 0.92 as main degradation products |
| $H_2O_2$ 0.3%, 1 h | 34% | 0.10; 0.16; 0.19; 0.50; 0.62; 0.92 |
| HCl 2 N, 2 h | 4% | 1.10 |
| Saturated carbonates, 1 h | 10% | 0.10; 0.12; 0.22; 0.39 |
| Light 48 h | 0% | – |

**Table 10. System Suitability Test for vitamin stability-indicating dosing methods.**

| Parameter | Thiamine hydrochloride | Riboflavin | Pyridoxine hydrochloride | Biotin |
|---|---|---|---|---|
| K' (capacity factor) | > 2[a] | > 2[a] | > 2[a] | > 2[a] |
| Rs (Resolution) | > 4[b] | > 2[a] | > 2.5[d] | > 2[a] |
| N (theoretical plates) | > 1,500[b] | > 2,000[a] | > 2,000[a] | > 2,000[a] |
| T (Tailing Factor) | <2[b] | ≤ 2[a] | ≤ 2[a] | ≤ 1,5[e] |
| RSD (from 6 analyses) | 2%[c] | 2% | 2%[d] | 2%[e] |

[a] FDA recommendations

[b] Thiamine hydrochloride API monograph, USP

[c] Thiamine hydrochloride oral solution monograph, USP

[d] Pyridoxine hydrochloride API monograph, USP

[e] Biotin tablets monograph, USP

**Table 11. Vitamin hard gelatin capsules, batches released and rejected since implementation.**

| | Batches released | Batches rejected |
|---|---|---|
| Thiamine hydrochloride (protocol 1) | 11 | 8 |
| Thiamine hydrochloride (protocol 2) | 10 | 0 |
| Riboflavin | 13 | 1 |
| Pyridoxine hydrochloride | 14 | 1 |
| Biotin | 13 | 3 |

Thiamine hydrochloride 50 mg and riboflavin 50 mg hard gelatin capsules were implemented in our hospital in 2020, whereas pyridoxine hydrochloride 50 mg and biotin 40 mg production began in 2022. Routine quality control and results from the first three batches analyzed are summarized in the supporting information (S1 file). Moreover, in 2022, the compounding protocol of thiamin hydrochloride was modified from a volume-based method to a weight-based method, and silica was added as an excipient, because almost 40% of batches from our initial protocol were rejected during quality control, generally for failure of uniformity of dosage units test (acceptance value higher than 15.0%). The other formulas did not yield to a recurrent batch rejection (Table 11).

In hard gelatin capsules, thiamine hydrochloride (from protocol 1, volume-based method), riboflavin, pyridoxine hydrochloride, and biotin contents remained higher than 90% of their initial contents for at least 12 months when stored at 25 °C/ 60% RH. Moreover, thiamine and pyridoxine contents remained higher than 95% of their initial contents from

the same duration, whereas riboflavin and biotin contents were only higher than 95% for 6 months. Results are summarized in Fig 1. This figure is also divided into 4 figures, one for each molecule, with CI95 (represented as thin lines), available in the supporting information (S4 Fig).

The stability study conducted for thiamine hydrochloride made from protocol 2 (weight-based method and silica) evidenced lower stability, as thiamine content (+/- CI95) decreased under 90% of thiamine initial content after 12 months (Fig 2, CI95 represented as thin lines).

Therefore, thiamine hydrochloride 50 mg hard gelatin capsules from protocol 2 can be considered stable only for 6 months when stored at 25 °C/ 60% RH.

For thiamine hydrochloride, riboflavin and pyridoxine hydrochloride, no statistically significant decrease in active ingredient content over time was found during a 12-month period. For biotin capsules, a significant reduction in content was observed after 12 months (p = 0.002). This decrease was not statistically significant during the first 6 months (p = 0.082).

Degradation products were also checked along stability studies. For thiamine hydrochloride, one peak (Relative Retention Time, RRT 0.25) was found since time zero with a relative area value between 0.05% and 0.10%. A very slight increase in the relative area of this peak was observed during the two stability studies (protocol 1 and protocol 2), to attain approximatively 0.20% after 6 months and 0.40–0.50% after 12 months. Moreover, at 12 months, a second small peak appeared (RRT 0.32) with a relative area approximately equal to 0.10%. Both for riboflavin and biotin, the impurity profile did not differ from time zero to 12 months after storage at 25 °C/ 60% RH. For pyridoxine hydrochloride, three new very small peaks (relative areas < 0.01%) were found at respectively RRT 1.27, 1.52, and 1.91. These peaks could be attributed to degradation products, but their relative areas are too small to be considered.

Both disintegration tests, TAMC and TYMC remained compliant throughout the study for each of the vitamins tested.

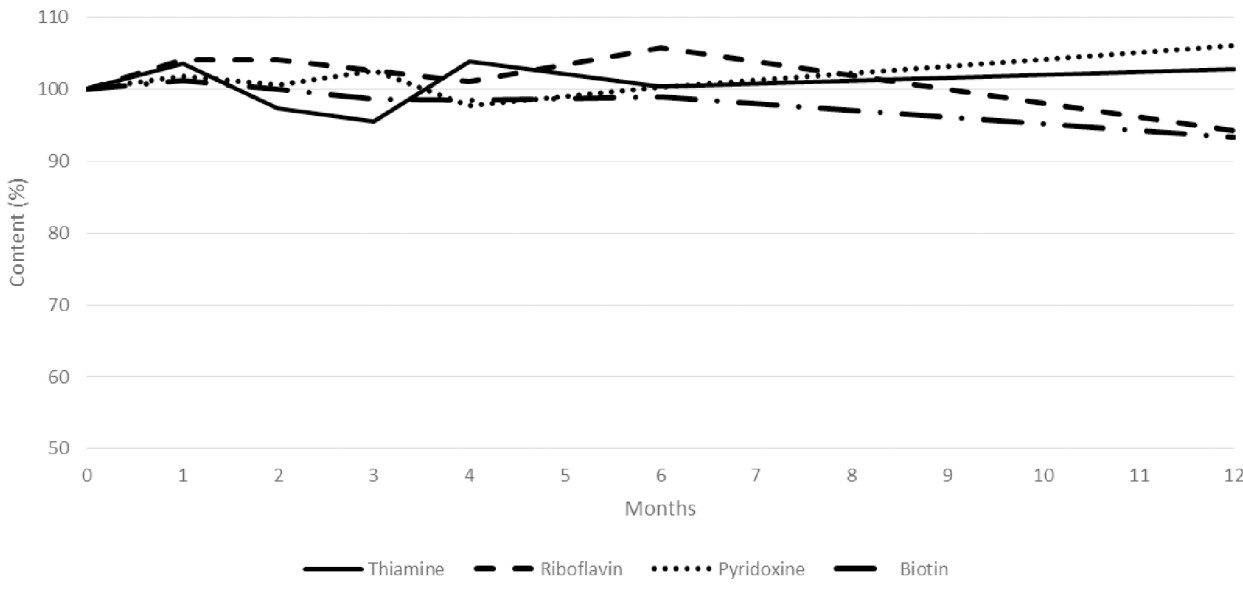

**Fig 1. Content variation of vitamin capsules stored at ambient temperature (25 °C/ 60% RH) over time.**

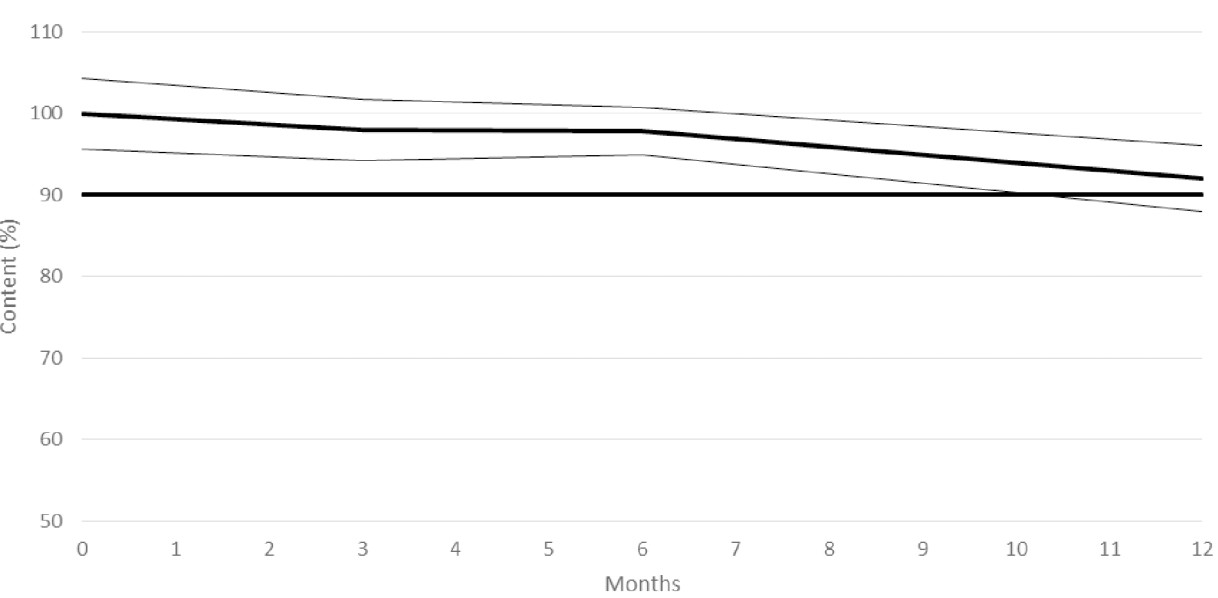

**Fig 2. Content variation of thiamine hydrochloride 50 mg capsules from protocol 2 stored at ambient temperature (25 °C/ 60% RH) over time.**

## 4. Discussion

Chromatographic methods for thiamine hydrochloride, riboflavin, pyridoxine hydrochloride, and biotin were all validated and proved to be stability-indicating. The absence of interference between the Active Pharmaceutical Ingredients (APIs) and their degradation products identified during forced degradation studies showed that our protocol can be used to conduct a stability study. Some industrial protocols also analyze a mixture of the API and its known degradation products, but it is not mandatory [28–29].

Our results show that riboflavin is very sensitive to light exposure (64% degradation after 20 min) and yields many degradation products, with one main peak identified at RRT 3.76. According to previous works, this peak could be attributed to the N-dealkylated product, named lumichrome [30–31]. Riboflavin also appeared sensitive to alkaline hydrolysis, yielding two degradation products (RRT 1.37 and 1.82). In the literature, alkaline degradation products of riboflavin were described as a quinoxaline derivative having pharmacological hypotensive properties [32] and as a β-ketoacid [31].

Under our experimental conditions, pyridoxine hydrochloride was also very sensitive to light exposure (35% degradation after 96 h) and moderately sensitive to oxidation (11% degradation after 96 h with $H_2O_2$ 15%). However, pyridoxine hydrochloride was resistant to degradation under heat, acid or basic conditions. Our results are similar to an old study [33], where pyridoxine was quickly degraded under artificial or natural light irradiation. In this study, the authors described a 44% degradation rate under oxidative condition ($H_2O_2$ 3%) but the absence of degradation after 1h in boiling water or after 1h in HCl 5 N at 100 °C or NaOH 5 N at 100 °C. Only nitric acid 2 N at 100 °C degraded pyridoxine (22% degradation rate) after 1h. These results confirm our results, as they are very close together. Another study found results quite similar to ours [34], with an important light degradation of pyridoxine and a moderate degradation under oxidative conditions (20% degradation rate, $H_2O_2$ 3% after 30 min). However, the authors also noticed degradation under acid/heat (18%, HCl 1 N after 1 h at 80 °C) and basic/heat (10%, NaOH 1 N after 1 h at 80 °C). A third study also described forced degradation experiments

on pyridoxine (acid, basic, and oxidative conditions), but unfortunately, the authors did not describe detailed results [35].

Under our experimental conditions, biotin was also found to be sensitive to oxidation as 34% were degraded after 1 h under moderate oxidative conditions ($H_2O_2$ 0.3%). As biotin bears a sulfur atom, its oxidation is unsurprising, as it can be easily oxidized to biotin sulfoxide or biotin sulfone [36]. It was also very sensitive to alkaline conditions, as NaOH trials always yielded total degradation. Finally with a saturated carbonates solution, only 10% degradation was found after 1 h. Biotin was also found to be relatively resistant to acid and light degradation. In the literature, from our knowledge, forced degradation experiments were never described on biotin before.

Then, vitamin hard gelatin capsules were compounded to have the simplest formulation, using only microcrystalline cellulose when possible. Silica was added to improve the powder flowability [37] before capsule filling only when this parameter was identified as a limitation during the initial batches compounding (thiamine hydrochloride and biotin).

Stability studies were conducted on vitamin hard gelatin capsules. From our experimental results, riboflavin 50 mg capsules, pyridoxine hydrochloride 50 mg capsules, and biotin 40 mg capsules are stable for one year when stored at 25 °C/ 60% RH. During the stability study on thiamine hydrochloride 50 mg capsules, degradation products were identified and increased over time. In the pharmaceutical industry, when degradation products are identified and characterized, their pharmacological relevance and/or their toxicity can be studied. However, during small-scale compounding's stability studies, hospital pharmacists do not have the resources to perform such studies. This is one of the limitations of our study, but also more generally of every published stability study, attempting that the pharmaceutical industry rarely publishes its data.

For thiamine hydrochloride active pharmaceutical ingredient batches released (not for stability study), European Pharmacopoeia imposes reporting thresholds at 0.05%, individual unspecified impurities maximal value not higher than 0.10% and total values for all impurities not higher than 0.50%. On the other hand, USP, for the same thiamine hydrochloride active pharmaceutical ingredient, only imposes a total value for all impurities that is not higher than 1.00%. With regards to these data and the ICH thresholds (reporting thresholds 0.05% and identification and qualification thresholds of 0.20% for a thiamine maximum daily dose of 1,500 mg.day$^{-1}$) we considered a conformity threshold of 0.20% for thiamine hydrochloride for our stability study.

Thiamine hydrochloride 50 mg hard gelatin capsules made from protocol 2 (microcrystalline cellulose/ silica as excipients) were stable for 6 months when stored at 25 °C/ 60% RH despite the appearance of several degradation products.

In routine practice, in our hospital, these capsules are solubilized in a small volume of water (generally 2 mL) to be administered. The whole volume can be administered, but sometimes, when a reduced dosage is required, the volume is only partially administered to correspond to the prescribed dosage. Such practice must require knowledge of the drug solubility, as if a drug is not very soluble in water, it can yield to a suspension. Therefore, sedimentation can occur, and the exact dosage will not be administered.

Thiamine hydrochloride 50 mg, and pyridoxine hydrochloride 50 mg have a good water solubility, respectively an approximate water solubility of 1000 mg.mL$^{-1}$ [38], and 220 mg.mL$^{-1}$ [39]. Therefore, when capsules are poured in 2 mL of water, a solution is obtained without any risk of underdosing.

In contrast, riboflavin and biotin are not as water-soluble, with respectively 0.08 mg.mL$^{-1}$ [40], and 0.22 mg.mL$^{-1}$ water-solubility values [41]. When capsules are opened in 2 mL of water, the resulting suspension should be gently mixed and fully administered to ensure proper dosing and avoid any risk of underdosing.

However, compounding hard gelatin capsules in advance ensures proper administration in pediatric practice. The definition and regulation of pharmaceutical compounding are not harmonized worldwide, but in several countries, two categories of preparations exist: "extemporaneous preparations" and "stock preparations" [42–43]. In the literature, there isno comparison of these two practices, but only "stock preparations" can be fully analytically controlled with destructive sampling [44]. In the present work, capsules are not made as emergency, protocols are pre-validated, and an independent quality control laboratory controls the resulting batches. Since their implementation, more than 10 batches of 300 capsules were stored and dispensed, without any report of undesirable event.

## 5. Conclusion

Thiamine hydrochloride 50 mg, riboflavin 50 mg, pyridoxine hydrochloride 50 mg, and biotin 40 mg hard gelatin capsules can be compounded as batches of 300 capsules by hospital pharmacists as treatments for pediatric metabolic diseases. Thiamine hydrochloride 50 mg hard gelatin capsules with microcrystalline cellulose and silica as excipients are stable for 6 months when stored at 25 ˚C/ 60% RH protected from light, whereas riboflavin 50 mg with microcrystalline cellulose, pyridoxine hydrochloride 50 mg with microcrystalline cellulose and biotin 40 mg with microcrystalline cellulose/silica are stable for one year when stored at 25 ˚C/ 60% RH protected from light. These results allow the compounding in advance of batches controlled, stored, and quickly dispensed in case of an emergency, such decreasing the risk of error and/or iatrogenic event.

## Supporting information

**S1 Fig. Thiamine hydrochloride representative chromatograms.**
(DOCX)

**S2 Fig. Riboflavine representative chromatograms.**
(DOCX)

**S3 Fig. Pyridoxine hydrochloride representative chromatograms.**
(DOCX)

**S4 Fig. Biotin representative chromatograms.**
(DOCX)

**S5 Fig. Content variations of vitamin capsules among time.**
(DOCX)

**S1 Table. Bulk and tapped densities of APIs and excipients.**
(DOCX)

**S1 File. Quality control for vitamin hard gelatin capsules.**
(DOCX)

## Author contributions

**Conceptualization:** Melanie Fuchs, Maeva Montaleytang, Christophe Curti.

**Data curation:** Pascal Rathelot.

**Formal analysis:** Romain Paoli-Lombardo.

**Investigation:** Marine Bonino, Amelie Bouchez, Amna Kadri, Ines Jacquet.

**Methodology:** Melanie Fuchs.

**Project administration:** Christophe Jean, Patrice Vanelle, Pascal Rathelot, Christophe Curti.

**Resources:** Christophe Jean, Thierry Terme.

**Supervision:** Romain Paoli-Lombardo, Patrice Vanelle.

**Validation:** Melanie Fuchs.

**Visualization:** Maeva Montaleytang, Thierry Terme.

**Writing – original draft:** Christophe Curti.

**Writing – review & editing:** Marine Bonino, Amelie Bouchez, Amna Kadri, Ines Jacquet, Romain Paoli-Lombardo, Christophe Jean, Melanie Fuchs, Maeva Montaleytang, Patrice Vanelle, Thierry Terme, Pascal Rathelot.

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
