## [Decision Letter · Decision Letter 0]

21 Jan 2025

PONE-D-24-56221Thiamine hydrochloride, riboflavin, pyridoxine hydrochloride, and biotin hard gelatin capsules prepared in advance and stored for the treatment of pediatric metabolic diseases: a safer alternativePLOS ONE

Dear Dr. Curti,

Thank you for submitting your manuscript to PLOS ONE. After careful consideration, we feel that it has merit but does not fully meet PLOS ONE’s publication criteria as it currently stands. Therefore, we invite you to submit a revised version of the manuscript that addresses the points raised during the review process.

We look forward to receiving your revised manuscript.

Kind regards,

Fırat Aşir

Academic Editor

PLOS ONE

Reviewers' comments:

Reviewer's Responses to Questions

**Comments to the Author**

1. Is the manuscript technically sound, and do the data support the conclusions?

Reviewer #1: Yes

Reviewer #2: Yes

Reviewer #3: Partly

2. Has the statistical analysis been performed appropriately and rigorously? 

Reviewer #1: No

Reviewer #2: N/A

Reviewer #3: N/A

3. Have the authors made all data underlying the findings in their manuscript fully available?

Reviewer #1: Yes

Reviewer #2: No

Reviewer #3: No

4. Is the manuscript presented in an intelligible fashion and written in standard English?

Reviewer #1: Yes

Reviewer #2: Yes

Reviewer #3: Yes

5. Review Comments to the Author

Reviewer #1: Dear Authors,

You conducted very important and applicable study for the pharmacy practice. However, I have some comments and questions:

line 26: I don't believe that magistral practice is characterized by the absence of any quality control, it's just not mandatory. Please correct

Please merge first two paragraph in the Introduction section

Please describe how to detect excipients with known effects in Table 1.

Methods: Why you used red carmine? Is it validated procedure for your API? Mixing one components do not mean mixing another one. Please provide information about mixing duration

Why silica din not used in the volume based method, as well as in riboflavine and pyridoxine capsules?

Why exactly 15 g of MCC was added? Do you have data about bulk and tapped density?

I suggest creating a Table with the formulations contents in the method section

line 167: Which water do you used?

Whether sinkers were used in the disintegration test?

line 220: I do not understand first sentence

Which guidelines you used for forced degradation studies?

Which statistical test was applied for conclusion about stability?

line 280: I do not understand what is dosing method in the context of HPLC

Please provide data about regulatory basis for compounding in advance

Reviewer #2: The paper demonstrates adequate scientific merit and professional value. However, some sections could benefit from improvements in grammatical expression to enhance clarity. I have outlined some specific observations in this regard below:

• Some results could be further interpreted. For instance, while degradation products of riboflavin are identified, their potential pharmacological relevance or toxicity is not addressed. Similarly, the implications of biotin degradation products, such as biotin sulfoxide, for the overall system are not explored.

• The text could benefit from more concise language. Some instances of grammatical errors and awkward phrasing, such as "yielded to 22% degradation," can be improved for better readability.

• Line 49 – 50. For clarity, consider revising the sentence “Although the parenteral route is often prescribed as an emergency treatment both for thiamine3 and pyridoxine, the oral route of administration must be privileged as soon as possible.” to “While the parenteral route is commonly used for emergency treatment with thiamine and pyridoxine, the oral route should be prioritized as soon as feasible.”

• Line 105, insert the word “rate” between the words “at” and “a” in the fragment “…(VWR)) at a flow of 1 mL.min-1”

• Line 112. Replace the word “weighted with “weighed”

• Line 126. Replace the word “weighted with “weighed”

• Line 138 Replace the word “weighted with “weighed”

• Line 158 Replace the word “weighted with “weighed”

• This problem persists through the methods – correct all

• Line 354 – 356. For better clarity, rewrite the sentence “When capsules are poured in 2 mL of water, the obtained suspension must be fully administered and gently mixed when it is taken before administering to avoid any risk of underdosing.” The fragment “.. the obtained suspension must be fully administered and gently mixed when it is taken before administering to avoid any risk of underdosing” is difficult to understand. A suggested version is “When capsules are dissolved in 2 mL of water, the resulting suspension should be gently mixed and fully administered to ensure proper dosing and avoid any risk of underdosing.”

• Consider revising the sentence “However, hard gelatin capsules compounding in advance secures the adminsttration in pediatric practice.” in line 357 to read as: "However, compounding hard gelatin capsules in advance ensures proper administration in pediatric practice."

Reviewer #3: This article by M. Bonino and colleagues presents a scientific investigation into the stability of four compounded hard capsules, containing essential vitamins, either thiamine hydrochloride (50 mg), riboflavin (50 mg), pyridoxine hydrochloride (50 mg), or biotin (40 mg). This article is particularly relevant to the hospital pharmacy community as it provides valuable data on the stability profiles of these combinations when compounded into capsule form. The findings of this study could have important practical applications for pharmacists involved in preparing and dispensing compounded medications for pediatric patients with specific vitamin deficiencies or related conditions.

The manuscript is well-written, scientifically sound, and presents a methodologically rigorous examination of the stability of these compounded capsules. The authors’ approach is thorough, and the results provide useful insights that could help hospital pharmacists ensure the safe and effective use of these vitamins.

A significant flaw in the manuscript is the lack of results regarding the 'uniformity of dosage units' assay, which is a critical test as per the European and US Pharmacopeia guidelines. Uniformity of dosage units is essential to ensure that each capsule contains the correct amount of active ingredient. Although it is likely that the authors performed this assay, the manuscript does not present any data on weight uniformity or a full pharmacopeial assay on drug content.

This omission is a major concern because without this data, it is difficult to assess whether the observed variability in the stability results stems from inconsistencies in the compounding process itself or from factors unrelated to formulation stability. In the absence of dosage uniformity data, the article lacks the necessary information to fully evaluate the consistency of the compounded capsules, which could ultimately affect their safety and efficacy in a clinical setting.

Other points to improve:

Introduction section:

1- Neonates and Hard Capsules: The authors mention neonates in the context of the study, but hard capsules are generally not considered an appropriate dosage form for this age group due to issues such as swallowing difficulties and the need for precise dosing. It would be helpful if the authors could provide further explanation or justification for why hard capsules need to be compounded in this case, particularly for neonates. Are there specific circumstances or clinical reasons that warrant this formulation, or could alternative dosage forms (e.g., liquids or powders) be more suitable for this population?

2- Table 1 - Terminology: In Table 1, the term ‘commercial drugs’ could be more accurately described as ‘available commercial formulations’ or ‘available commercial drug products’. This would provide a clearer and more precise reference. Additionally, it would be beneficial to specify which country or region this information pertains to, as pharmaceutical formulations can vary significantly by location.

3- Scope of the Article: The statement ‘In our hospital, prescribers…’ refers to a very specific context, which may limit the broader applicability of the article's findings. It would be helpful if the authors could clarify that the scope of the study extends beyond this specific setting. Ideally, the manuscript should aim to describe a problem or solution that is of general relevance to the wider hospital pharmacy community, rather than being confined to a single institutional context.

Methodology section:

4- Capsules Compounding: Please provide the exact weights of the powders used in the study, including the number of decimals that reflect the precision of the weighing scale. This will ensure greater transparency and reproducibility of the compounding process. Additionally, this section could be streamlined by describing the general compounding procedure briefly, followed by a table that presents the specific weightings for each formulation. This would make the methodology more concise and easier to follow.

5- HPLC Stability-Indicating Method: The description of the HPLC method could be more concise. This streamlined information will provide a clearer and more focused explanation without unnecessary repetition. Please include the following key details:

(i) The equipment used (e.g., HPLC system, detector, etc).

(ii) The exact duration of the analytical steps (sample analysis).

(iii) The duration of vortexing during sample preparation, as this can affect consistency.

6- Stability Studies: Please specify the reference for the climatic chamber used in the study. It would also be beneficial to use the terms ‘stability chamber’ or ‘environmental test chamber’ instead, as these are more commonly used in the pharmaceutical field to describe equipment used for stability testing. This terminology is more precise and aligns with established industry standards.

Results section:

7- Unless mistaken, this manuscript nor its supporting data does not seem to include the spectra for each compound alone and not degraded (reference spectra). This is a major concern as the readers should be able to judge how the degradation process affects the spectra.

8- Tables 3-4 : some RRT are close enough to the parent peak. Which guidelines or scientifically established data/norms did the authors use to validate the suitability of their methods in this matter ?

9- Table 7: include precise information on which criterion was accepted based on European or US pharmacopeia and why.

10- Table 8 and corresponding text: please precise which criterion/assay was used to reject the batches.

11- I would rather be able to see the standard deviations on each point, so maybe Figure 1 could be divided into 4 smaller figures to include this important dataset.

12- Figure 2: legend is not detailed enough to understand what is presented on this figure. Thin lines, thick lines? SD?

Discussion Section:

13- The discussion does not address whether the HPLC methods used are novel or adapted from existing protocols, such as those outlined in pharmacopeias or previous studies. Additionally, while degradation products are mentioned, they are only observed rather than identified.

14- For some of these products, the relative retention times (RRT) are similar to that of the parent drug, suggesting that the resolution of the method could be optimized. If improving the resolution is not feasible, this limitation should be explicitly acknowledged and discussed.

15- The discussion section is relatively lengthy and could benefit from being more concise while ensuring the relevance of each point.

16- Given that the results section requires revision, the discussion should also be adjusted to reflect these changes.

6. PLOS authors have the option to publish the peer review history of their article (what does this mean? ). If published, this will include your full peer review and any attached files.

**Do you want your identity to be public for this peer review?** For information about this choice, including consent withdrawal, please see our Privacy Policy .

Reviewer #1: No

Reviewer #2: No

Reviewer #3: No

---

## [Author Response · Author response to Decision Letter 1]

17 Feb 2025

Dear Editor and reviewers,

Thank you for the evaluation of our work. Please, find below a rebuttal letter that responds to each point raised by the reviewers.

Reviewer #1:

Line 26: I don't believe that magistral practice is characterized by the absence of any quality control, it's just not mandatory. Please correct

The sentence “, without any quality control,” was deleted

Please merge first two paragraph in the Introduction section

The first two paragraphs were merged.

Please describe how to detect excipients with known effects in Table 1.

While most excipients are considered inactive, some can have a known action or effect. In Europe, they must be declared on the label and package leaflet of the medicine. The EMA has established a list of “excipients with known effects”, which is very useful in compounding practice for pediatric patients.

L61. “can contain excipients with known effects” was replaced by “can contain excipients with known effects (as defined and listed by the European Medicine Agency13)”

A new reference was added:

European Medicine Agency. Excipients labelling. [Cited 2025 January 27]. Available from: https://www.ema.europa.eu/en/human-regulatory-overview/marketing-authorisation/product-information-requirements/excipients-labelling

Methods: Why you used red carmine? Is it validated procedure for your API? Mixing one components do not mean mixing another one. Please provide information about mixing duration

We thank the reviewer for this remark. We omitted the mixing duration, which is longer for thiamine (25 min) than for others (10 min).

A table (Table 2) and a paragraph summarizing the general compounding procedure (including mixing duration) replaced the text from L75 to L99

Red carmine is routinely used as a tracer dye, as a complement visual help (“homogenization tracer”), to ensure powder uniformity. We totally agree with the reviewer’s comment (Mixing one component does not mean mixing another one), but red carmine is only complementary to validated mixing times.

Why silica din not used in the volume based method, as well as in riboflavine and pyridoxine capsules?

In the discussion, L315, we added a short explanation paragraph:

“Then, vitamin hard gelatin capsules were compounded to have the simplest formulation, using only microcrystalline cellulose when possible. Silica was added to improve the powder flowability32 before capsule filling only when this parameter was identified as a limitation during the initial batches compounding (thiamine hydrochloride and biotin).”

One reference was added:

Majerová D, Kulaviak L, Růžička M, Štěpánek F, Zámostný P. Effect of colloidal silica on rheological properties of common pharmaceutical excipients. Eur J Pharm Biopharm. 2016; 106: 2-8.

Why exactly 15 g of MCC was added? Do you have data about bulk and tapped density?

Yes, the weight values were calculated from bulk and tapped densities. We obtained decimal values, but decided to slightly round these values (with a difference between the theoretical and rounded weights lower than 1%, but with the correct amount of API). For example, for pyridoxine capsules, the theoretical weight is 27.09 g, rounded to 27.00 g (15.00 g of pyridoxine and 12.00 g of cellulose). There is a difference of approximately 0.3% between the theoretical weight (from tapped density) and the rounded weight.

We had not added these data in the manuscript to not extend it too much, but it was a mistake as two reviewers made this remark. We made the following corrections:

“To determine the formula for each vitamin capsule, we first evaluated the bulk and tapped densities for each individual component and for selected mixtures, using the same protocol as previously described.17 Then, we calculated the theoretical weights to achieve a volume of 63 mL (300 hard gelatin capsules, size 4) with the appropriate amount of Active Pharmaceutical Ingredient (API). Finally, we slightly rounded these theoretical weights to obtain a simplified formula, ensuring differences between the theoretical weight and the rounded weight lower than 1%. The results are summarized in a table in the supporting information.”

A new reference “Wasilewski M, Curti C, Bouguergour C, Panuccio C, Thevin P, Primas N, Lamy E, Vanelle P. Paediatric capsule compounding in hospital practices: by weight or by volume? Eur J Hosp Pharm. 2023; 30: 363-366.” was added

“Table A. Bulk and tapped densities of APIs and excipients” was also added in Supporting information.

I suggest creating a Table with the formulations contents in the method section

A table (Table 2) and a paragraph summarizing the general compounding procedure replaced the text from L75 to L99

Line 167: Which water do you used?

L167: “(distilled water, Cooper)” was added

Whether sinkers were used in the disintegration test?

No, we sometimes use sinkers for the dissolution test but never for disintegration. For disintegration test, disks can be used as per USP, but we didn’t use them for the tests described in this work.

L167: “and without disks” was added

Line 220: I do not understand first sentence

We apologize for the lack of clarity.

L220: “Thiamine hydrochloride dosing method was found to be linear between 50 and 500 µg.mL-1, riboflavin between 100 and 200 µg.mL-1, pyridoxine hydrochloride between 200 and 800 µg.mL-1 and biotin between 100 and 500 µg.mL-1” was replaced by

“Linearity was demonstrated for all four methods: between 50 and 500 µg.mL-1 for thiamine hydrochloride, between 100 and 200 µg.mL-1 for riboflavin, between 200 and 800 µg.mL-1 for pyridoxine hydrochloride and between 100 and 500 µg.mL-1 for biotin.”

Which guidelines you used for forced degradation studies?

Protocols for forced degradation studies are not detailed in the ICH. However, we used Methodological guidelines for stability studies of hospital pharmaceutical preparations established by a French society (GERPAC). These guidelines are very close to the protocols and requirements described in two frequently cited publications on the subject:

Blessy M, Patel RD, Prajapati PN, Agrawal YK. Development of forced degradation and stability indicating studies of drugs-A review. J Pharm Anal. 2014; 4: 159-165.

Bakshi M, Singh S. Development of validated stability-indicating assay methods--critical review. J Pharm Biomed Anal. 2002; 28: 1011-1040.

We already cited these publications in our manuscript (L284)

Which statistical test was applied for conclusion about stability?

As suggested in “ICH Topic Q 1 E Evaluation of Stability Data”, a 95 percent confidence limit was calculated and used to determine stability.

“An appropriate approach to retest period or shelf life estimation is to analyze a quantitative attribute (e.g., assay, degradation products) by determining the earliest time at which the 95 percent confidence limit for the mean intersects the proposed acceptance criterion.”

However, to answer the reviewer’s request, we realized a statistical analysis (to study batch poolability and the statistical significance of drug’s content variations), and the results were added to the manuscript:

L218: “The statistical analysis of the change in measured content over time was conducted in accordance with the ICH Q1E guidelines. Specifically, batch poolability was confirmed through an analysis of covariance (ANCOVA) using a significance level of 0.25. Since time × batch interactions and the main effect of batch were non-significant for each of the formulations studied, all batches were deemed poolable. Then, the relationship between the variation in active ingredient content over 12 months and time was estimated for each formulation using regression analysis and the calculation of the coefficient of determination R2.” was added

L266: “For thiamine hydrochloride, riboflavin and pyridoxine hydrochloride, no statistically significant decrease in active ingredient content over time was found during a 12-month period. For biotin capsules, a significant reduction in content was observed after 12 months (p = 0.002). This decrease was not statistically significant during the first 6 months (p = 0.082).” was added

Line 280: I do not understand what is dosing method in the context of HPLC

L280 (and L30): “dosing” was replaced by “chromatographic”

Please provide data about regulatory basis for compounding in advance

In our country, compounding in advance is allowed (“hospital preparations”, versus “magistral preparations” which are extemporaneously compounded) and involves analytical quality control. As this quality control decreases the risk of error, hospital pharmacists try when possible to favor such “standardized” preparations. Moreover, as there was several compounding errors in pediatric practice due to extemporaneous compounding without control, our regulatory agencies encourage us to realize “standardized” and controlled preparations. However, as the definition and regulation of pharmaceutical compounding are not harmonized worldwide, we slightly modified our manuscript (without detailing our national regulatory rules).

L357: “Hard gelatin capsules compounding in advance secures the administration in pediatric practice. These capsules are not made as emergency” was replaced by: “However, compounding hard gelatin capsules in advance ensures proper administration in pediatric practice. The definition and regulation of pharmaceutical compounding are not harmonized worldwide, but in several countries, two categories of preparations exist: “extemporaneous preparations” and “stock preparations”.41-42 In the literature, there is no comparison of these two practices, but only “stock preparations” can be fully analytically controlled with destructive sampling.43 In the present work, capsules are not made as emergency”

Three references were added:

41. Dooms M, Carvalho M. Compounded medication for patients with rare diseases. Orphanet J Rare Dis. 2018; 13: 1.

42. Carvalho M, Almeida IF. The Role of Pharmaceutical Compounding in Promoting Medication Adherence. Pharmaceuticals (Basel). 2022; 15: 1091.

43. Uriel M, Marro D, Gómez Rincón C. An Adequate Pharmaceutical Quality System for Personalized Preparation. Pharmaceutics. 2023; 15: 800.

Reviewer #2:

• Some results could be further interpreted. For instance, while degradation products of riboflavin are identified, their potential pharmacological relevance or toxicity is not addressed. Similarly, the implications of biotin degradation products, such as biotin sulfoxide, for the overall system are not explored.

We agree with this remark and wish could do it, but public hospital structures do not have the resources for such studies. We added this point as a limitation of our work in the discussion.

L319: “In the pharmaceutical industry, when degradation products are identified and characterized, their pharmacological relevance and/or their toxicity can be studied. However, during small-scale compounding’s stability studies, hospital pharmacists do not have the resources to perform such studies. This is one of the limitations of our study, but also more generally of every published stability study, attempting that the pharmaceutical industry rarely publishes its data.” was added.

• The text could benefit from more concise language. Some instances of grammatical errors and awkward phrasing, such as "yielded to 22% degradation," can be improved for better readability.

L300: “yielded to 22% degradation” was modified to “degraded pyridoxine (22% degradation rate)”.

Several slight modifications and table adjunctions were done to improve readability.

• Line 49 – 50. For clarity, consider revising the sentence “Although the parenteral route is often prescribed as an emergency treatment both for thiamine3 and pyridoxine, the oral route of administration must be privileged as soon as possible.” to “While the parenteral route is commonly used for emergency treatment with thiamine and pyridoxine, the oral route should be prioritized as soon as feasible.”

L49-50: “Although the parenteral … must be privileged as soon as possible.” Was replaced by “While the parenteral route is commonly used for emergency treatment with thiamine and pyridoxine, the oral route should be prioritized as soon as feasible.”

• Line 105, insert the word “rate” between the words “at” and “a” in the fragment “…(VWR)) at a flow of 1 mL.min-1”

These sentences were deleted and replaced by a synthetic table

• Line 112. Replace the word “weighted with “weighed”

• This problem persists through the methods – correct all

“weighted” was replaced by “weighed” in the whole manuscript

• Line 354 – 356. For better clarity, rewrite the sentence “When capsules are poured in 2 mL of water, the obtained suspension must be fully administered and gently mixed when it is taken before administering to avoid any risk of underdosing.” The fragment “.. the obtained suspension must be fully administered and gently mixed when it is taken before administering to avoid any risk of underdosing” is difficult to understand. A suggested version is “When capsules are dissolved in 2 mL of water, the resulting suspension should be gently mixed and fully administered to ensure proper dosing and avoid any risk of underdosing.”

L354: “When capsules are poured in 2 mL of water, the obtained suspension must be fully administered and gently mixed when it is taken before administering to avoid any risk of underdosing.” Was replaced by “When capsules are opened in 2 mL of water, the resulting suspension should be gently mixed and fully administered to ensure proper dosing and avoid any risk of underdosing.”

• Consider revising the sentence “However, hard gelatin capsules compounding in advance secures the administration in pediatric practice.” in line 357 to read as: "However, compounding hard gelatin capsules in advance ensures proper administration in pediatric practice."

L357: “hard gelatin capsules compounding in advance secures the administration in pediatric practice. These capsules are not made as emergency” was replaced by: “However, compounding hard gelatin capsules in advance ensures proper administration in pediatric practice.”

Reviewer #3:

A significant flaw in the manuscript is the lack of results regarding the 'uniformity of dosage units' assay, which is a critical test as per the European and US Pharmacopeia guidelines. Uniformity of dosage units is essential to ensure that each capsule contains the correct amount of active ingredient. Although it is likely that the authors performed this assay, the manuscript does not present any data on weight uniformity or a full pharmacopeial assay on drug content.

This omission is a major concern because without this data, it is difficult to assess whether the observed variability in the stability results stems from inconsistencies in the compounding process itself or from factors unrelated to formulation stability. In the absence of dosage uniformity data, the article lacks the necessary information to fully evaluate the consistency of the compounded capsules, which could ultimately affect their safety and efficacy in a clinical setting.

We did not include these results in our manuscript to decrease its length. It was a mistake.

All our hospital preparations are analytically and microbiologically controlled for the following parameters:

- Content

- Content uniformity. For capsules with a dose or a ratio drug/substance higher than 25 mg/25%, uniformity of dosage units (weigh variation test) is applied as per both EP and USP. One capsule is analyzed for its content, and 10 are accurately weighed individually, emptied and the shells are accurately weighed individually. The acceptance value is calculated and has to be lower than 15%.

- Microbiological contamination (TAMC and TYMC)

These data were summarized and added to the supporting Information. A table with the content and uniformity of dosage unit results from the first three batches released was also added to answer to reviewer’s request.

L244: “Routine quality control and results from the first three batches analyzed are summarized in the supporting information.” was added.

Introduction section:

1- Neonates and Hard Capsules: The authors mention neonates in the context of the study, but hard capsules are generally not considered

---

## [Decision Letter · Decision Letter 1]

3 Mar 2025

Thiamine hydrochloride, riboflavin, pyridoxine hydrochloride, and biotin hard gelatin capsules prepared in advance and stored for the treatment of pediatric metabolic diseases: a safer alternative

PONE-D-24-56221R1

Dear Dr. Curti,

We’re pleased to inform you that your manuscript has been judged scientifically suitable for publication and will be formally accepted for publication once it meets all outstanding technical requirements.

Kind regards,

Fırat Aşir

Academic Editor

PLOS ONE

Additional Editor Comments (optional):

Reviewers' comments:

Reviewer's Responses to Questions

**Comments to the Author**

1. If the authors have adequately addressed your comments raised in a previous round of review and you feel that this manuscript is now acceptable for publication, you may indicate that here to bypass the “Comments to the Author” section, enter your conflict of interest statement in the “Confidential to Editor” section, and submit your "Accept" recommendation.

Reviewer #1: All comments have been addressed

Reviewer #3: All comments have been addressed

2. Is the manuscript technically sound, and do the data support the conclusions?

Reviewer #1: Yes

Reviewer #3: Yes

3. Has the statistical analysis been performed appropriately and rigorously? 

Reviewer #1: Yes

Reviewer #3: Yes

4. Have the authors made all data underlying the findings in their manuscript fully available?

Reviewer #1: Yes

Reviewer #3: Yes

5. Is the manuscript presented in an intelligible fashion and written in standard English?

Reviewer #1: Yes

Reviewer #3: Yes

6. Review Comments to the Author

Reviewer #1: Language is clear. All is well written and explained. I do advice authors to read text one more time since some typos might be omitted.

Reviewer #3: (No Response)

7. PLOS authors have the option to publish the peer review history of their article (what does this mean? ). If published, this will include your full peer review and any attached files.

**Do you want your identity to be public for this peer review?** For information about this choice, including consent withdrawal, please see our Privacy Policy .

Reviewer #1: No

Reviewer #3: No

---

## [Editor Report · Acceptance letter]

PONE-D-24-56221R1

PLOS ONE

Dear Dr. Curti,

I'm pleased to inform you that your manuscript has been deemed suitable for publication in PLOS ONE. Congratulations! Your manuscript is now being handed over to our production team.

Kind regards,

on behalf of

Dr. PLOS Manuscript Reassignment

Staff Editor

PLOS ONE